# Afrocentric Education for Liberation in the Classroom: It Takes a Village to Raise a Child

**Tytianna Nikia Maria Ringstaff**

Digital Learning and Innovation, Simmons College of Kentucky, Louisville, KY 40203, USA; tringstaff@simmonscollegeky.edu

**Abstract:** Racist and inequitable schools in the United States espouse an anti-Black and color blind curriculum that negatively impacts Black students' lives. Black schools, including homeschools, are a strategic response to racist public and private schools and a viable option to address the academic and cultural needs of Black students. This paper explores Afrocentric practices, including familial relationships through culturally responsive instructional practices: an African time orientation, a personalized learning plan, authoritative teaching, *OurStory*, and Rising Meeting. This paper provides evidence that these practices benefit Black students. I draw upon an Afrocentric theoretical framework in this qualitative study to analyze and interpret data collected at the Black Scholars Academy (BSA), a pre-kindergarten (Pre-K) through 12th-grade Black homeschool collective in the U.S. The data consist of classroom observations, individual interviews with current and former teachers and students, and textual artifacts collected between July and November 2019. Familial relationships helped students develop cultural pride, agency, self-determination, independence, and liberation through education. The employment of Afrocentricity as a best practice in a homeschool collective is considered advisable across every educational context. There is a need for more research on Afrocentric practices as one of many culturally responsive techniques to best teach culturally diverse students, especially Black students, in educational settings.

**Keywords:** Afrocentricity; Afrocentric education; homeschooling; family; familial relationships; village; Afrocentric features; classroom; liberation

A people without knowledge of their past history, origin, and culture is like a tree without roots.—Marcus Garvey

## 1. Introduction

During the global pandemic of 2020 and 2021, trauma plagued our world. Post COVID-19 and in an era of the Black Lives Matter protests, critical race theory debates, and the 1619 discussions, Black families have taken up arms in many ways as they always have with one way of being, educating their children from a Black cultural perspective or from an Afrocentric perspective that places Black culture at the center of their curriculum and instruction. In independent Black schools, whether public, private, or homeschools, students are learning in the way of their culture, but not just any culture—Black culture. Historically, Black families have sought and continue to educate their children in race-centered and protection-oriented ways whether their children attended an American public or private school or were homeschooled.

This article acknowledges a few of the problems with American schools and recommends the implementation of Afrocentric curricula and instructional practices—specifically, familial relationships—found in Black homeschools to teachers and leaders in these schools due to their benefits to the academic and cultural development of all students, especially Black students.

As a response, homeschooling has become a viable option to address the academic and cultural needs of Black students [1–13]. In 2015, there were nearly 2.3 million school-

aged children entering homeschools across the country [13]. In 2015, out of that population, there were 220,000 Black homeschool families making the Black community the second largest racial group of homeschoolers [13–17].

Limited research exists on Black homeschools, especially Afrocentric homeschool collectives. Dr. Brian D. Ray, the founder of the National Home Education Research Institute (NHERI) which includes an academic journal and annual conference on home education, has researched homeschooling for over 30 years. Ray has written and published the most research on homeschoolers, deeming him as the lead researcher on homeschooling in the United States. His research primarily focuses on White Southern and religious homeschool families. However, Ray has also conducted research on the academic trends among Black and White homeschool students. His research has revealed positive trends among Black homeschoolers with a focus on academic performance that is not comprehensive or inclusive of larger research participant populations in the United States [8,10,13–19]. Ray's research certainly breaks ground in documenting the academic outcomes of Black homeschooling but fails to establish causation and explore race and culture.

Dr. Ama Mazama, formerly known as Marie-Josée Céro, is one of the leading Afrocentric scholars on Black homeschoolers in the United States, who has criticized Ray's study as racially unrepresentative of the homeschool population since it did not focus on Black homeschoolers. In response, in 2012, Mazama and Lundy [16] conducted a small study of 74 Black homeschool families throughout the United States to draw more conclusive evidence of the motivations behind why Black families homeschool. According to Mazama and Musumunu's research in 2014 [18], Black families homeschool their children for particular reasons that include the following from the most prevalent to the least prevalent: (a) concerns about the environment of other schools; (b) moral instruction; and (c) dissatisfaction with academic instruction at other schools. Black parents are dissatisfied with academic instruction in public schools for the following reasons: (a) preservice teachers are often ill-prepared; (b) the presence of Black teachers in schools continues to decrease; and (c) the curriculum and instruction is largely White and culturally exclusionary [19,20]. However, the primary reason that Black families homeschool their children is to protect them from racism [13,14,19–23]. Black families are dissatisfied with public and private schools which is the chief reason they are seeking to rectify the omissions of Black identity and culture prevalent in public and private schools [1,2,14,19]. Black families are homeschooling their children as a form of *racial protectionism*, a term coined by Ama Mazama, as a method of racial security. According to Mazama and Lundy [16], "Racial protectionists shared the view that schools, public or private, could not, given the racist nature of American society, be emotionally safe for Black children" [16] (p. 734).

Racism, including the high rates of the unjust killings of countless unarmed Black men and women at the hands of the fraternal order of police, is the primary determinant in Black parents' decision to homeschool their children. The goal is to protect them from racism in the larger society and schools. Protecting Black students from racism, a form of violence in school, is especially important as public schools, with a predominantly White staff, tend to punish Black students more harshly than White students and students are expected to engage in an anti-Black curriculum where high-stakes testing and rigid curricular demands negatively impact the academic achievement and cultural development of Black students, especially Black males, which has implications for widening the school-to-prison pipeline [13,15,16,18,19,21,24–29]. Thus, Black parents are protecting Black children from being placed in special education and suspended from school due to cultural misunderstandings that have existed between White teachers and Black students [13,27].

## 2. Examining the Miseducation and Criminalization of the Black Student

Exactly 65 years post *Brown v. Board of Education*, educational problems, developed from segregated schooling, integration, and resegregation, continue to play a role in the disenfranchisement of Black students in public and private schools [30].

Historically, American public and private schools have been led by predominantly White teachers and administrators, while Black students are the largest school-age minority population in public and private schools which has negatively impacted Black students' lives. These schools are often celebrated for their high-stakes testing, a narrow and anti-Black curriculum, and suspending Black students. These schools often promote the miseducation and criminalization of students. Public and private schools that espouse a racist curriculum, taught by a predominantly White teacher force, are miseducating and culturally disconnected from Black students' homes, lives, and literacies [20,31–34]. As a result, according to Crotty-Nicholson et al. [34], Black students are eight times more likely to be disciplined than their White counterparts. Nationally, Black students represent 18% of the population yet account for 48% of all school suspensions and expulsions and are ultimately funneled into the school-to-prison pipeline [14,17–19,22,23,26,35–39].

As an advocate for quality and equitable education for children, whether they attend public, private, or homeschools, I have found in my research that many curricula and instructional methods present in Black homeschools are beneficial not only to Black children in the homeschool environment but all children in the American public and private sectors where a culturally diverse student population continues to grow. While there are many successful methods to bring into American schools, this article focuses on the features and benefits of familial relationships, a component of Afrocentricity, that leads to a village mentality in a school environment. A familial relationship or village orientation between students and teachers has been shown to improve students' academic progress and cultural pride. This Afrocentric paradigm extends success beyond being academic to being cultural and communal, allowing students to engage in developmental learning opportunities with culture at the center of the communal experience of the classroom. Students are not only encouraged to succeed developmentally but to cultivate a cultural identity of pride and appreciation for one's African heritage through celebration in a communal environment that operates as a village or family. In Afrocentric schools, success is not measured only by academics but students' ability to connect to their cultural heritage within a community. As the African adage reveals, "It takes a village to raise a child".

### 3. Methodology

My interest in the topic of homeschools, particularly Black homeschools, began in 2013 with my growing clientele in children's books including Black homeschool families. During my first book vending experience, I sold the first volume of my children's book to an African American mother who was homeschooling her child. She was teaching her young son how to read and was searching for African American or multicultural books with culturally representative characters. After she purchased the book, I inquired about the mother's homeschooling experience. During that conversation, I learned that there were other African American homeschool families who she was connected with which led me to learn about Yala and her Black homeschool collective and schedule a preliminary phone call and interview with her which included a tour of the BSA in 2017.

In this article, I am drawing upon an Afrocentric theoretical framework as it informs my study of the Black homeschool collective where the teacher taught from an Afrocentric lens. An Afrocentric theoretical framework [40–42] provides a method of analysis for this study. Afrocentricity places Black people and their experiences at the center of phenomena. In the context of this study, Afrocentricity places Black students and their experiences at the center of the educational experience.

Afrocentricity, a concept coined during the 1960s civil rights era by Molefi Kete Asante, is the product of many scholars [31,43,44]. While the philosophical discussions of Afrocentricity, agreed upon or contested across the scholarly field, are not the focus of this chapter, it is imperative to recognize that there are advocates and critics of Afrocentricity. Kwame Anthony Appiah, an African scholar, is one who critiques Afrocentricity and argues that Afrocentricity does two things: (a) it inadvertently does precisely what Europe had done in talking of itself superlatively and trashing all non-European cultures and

(b) Afrocentric ideology ignores the realism of African peoples and experiences, which like all those other humans, contain both good and bad, true and false, etc., in their different understandings of the world [45].

Essentially, Afrocentricity is not anti-White but grounded in the legacy of Black culture and excellence [32–34,38]. Afrocentric education focuses on the experiences and perspectives of Black people at the core of its curriculum. It calls for the recentering of content that focuses on the Black experience, African perspective, and liberation of the Black person and community [39–49]. Afrocentric education is designed to liberate Black students who have been historically marginalized and disenfranchised through their indoctrination of European history and culture which has contributed to slavery, racial segregation, and discrimination. An Afrocentric curriculum consists of four key goals towards affirming Black students' lives, including the following: challenging racism and hegemony, providing differentiated learning styles, promoting a positive self-concept and collective identity among Black students, and a providing a model for multicultural education [46].

According to Shockley and Frederick (2010), "Afrocentric educationists attempt to offer methods, ideas, and concepts that are best suitable for reaching children of African descent" [50] (p. 1213). The goal of Afrocentric educationists is to create an education for Black students "where the focus could be on loving oneself and practicing one's own culture, focusing on oneself as the subject of history instead of the object of someone else's stories, accepting the anteriority of the early African civilizations, and attempting to construct a unified, pan-Africanist reality for people of African descent" [50] (p. 1221).

An Afrocentric theoretical framework was used to inform this study of the Black homeschool collective with respect to the homeschool teacher's style of teaching. As a self-identified Afrocentric teacher, this framework is most fitting to this study as it honors African spirituality and its conceptual ideas and values as identified with the use of the Kwanzaa principles as a curriculum and instructional strategy.

In an attempt to provide hope during an intense racial environment in 1966, Maulana Karenga invented the *Nguzo Saba* (meaning *The Seven Principles* in Kiswahili) and systemized it as Kwanzaa—a national African American holiday honoring and celebrating African heritage, culture, identity, family, and community [35,43–45,47–49].

While *Kwanzaa* is not African in tradition, its principles and definitions are African in origin [35,39,42,45]. Karenga created an extended interpretation of Kiswahili terms, principles, and definitions for the Black community. The Kiswahili language system "is a language with distinct genetic linkage to the languages of the northeast, central, and southern Africa" [47] (p. 144). Karenga intended to help build and reinforce unity or *umoja* among people of African ancestry as a system of seven non-denominational principles as a method of support [48].

Dr. Wade Nobles, an Afrocentrist and Black psychologist, instituted an African-centered curriculum with the goal to recenter, refine, recover, reproduce, and reclaim Black identity [46]. This is the spiritual calling of Afrocentricity and its many features.

The benefits and best practices of Afrocentric education allow all teachers who understand teaching from this framework to instruct in ways that support all students, especially Black students' development of agency, self-determination, and independence in a racially protective space. This article argues that familial relationships—the most salient theme in the data—are a component of Afrocentricity, the theoretical framework undergirding this study.

Grounded in the literature on Black education, is the phenomenon that many Black people have sought better opportunities for their children and were agentic, self-determined, and independent in the creation of independent Black schools as an alternative to racist American schools designed with only the White student in mind. In these schools, offering a quality and equitable education that promoted a familial or village-oriented environment was essential to teachers.

Familial relationships manifest in the national and local iterations of Black education, including homeschools. Family or village-oriented curricula and instruction found in many

Black homeschools are great examples for American public and private schools. A familial or village-oriented relationship in the school is a method of student intervention and racial protectionism.

While Afrocentric education embraces familial relationships, this framework is not the only cultural framework that values familial relationships. Familial relationships are found and different across all cultural groups, which should also be considered. Appreciation and affirmation are shared in various ways in different cultures that should be further examined as culturally appropriate and responsive to a diverse population of students. The teaching practices provided in this article serve as suggestions that can be used in any educational setting. However, it is not the only approach. Yala, the teacher, is referenced as a model for some of her Afrocentric practices due to the impact of how the development and presence of a familial relationship benefits her students academically and culturally.

While this article offers suggestions to American schools as per research on Black homeschools, limited research exists on Black homeschools, especially Afrocentric homeschool collectives. There is also not enough evidence to prove that the features of Afrocentricity present at the BSA inform and guide students on a long-term basis, beyond the limited time during which I conducted this study. This research is not a longitudinal study conducted over a lengthy period. Future studies are needed to investigate the impact of Afrocentric education as providing culturally responsive best practices to students and as best practices in any educational setting. The following includes the research questions that guide this article.

### Research Questions

1. What salient features are present in the curriculum and instruction at the Black Scholars Academy?
2. How were familial relationships beneficial to the Afrocentric vision, mission, and goals of this homeschool collective?

### 4. Methods

This qualitative study, conducted during the 2019–2020 school year, explores the Black Scholars Academy (BSA), a Black homeschool collective in the Midwest of the United States. Founded in 2013, the BSA is a pre-kindergarten (Pre-K) to 12th-grade homeschool collective. Data collected during this study consisted of six full days of classroom observations, individual interviews with the research participants for an in-depth analysis and interpretation of the research, and an evaluation of textual artifacts, including teacher materials, such as books and content on the collective's website.

Data were collected for four months from July 2019 to November 2019 at the BSA. During the 2019–2020 school year, the student and teacher population comprised one teacher and seven students ranging from 3 to 17 years old, all of whom identified as Black or African American except for one student who identified as "biracial" or as having two or more races (specifically White and Black) at the BSA. While this homeschool included students in early childhood to the 12th grade, this study focused only on the older group of students in the 5th through 12th grades.

I interviewed one teacher and four current students and transcribed six classroom observations at the BSA. This research was conducted in five phases, emphasizing the following research methods and procedures: (a) communication for the teacher interview; (b) teacher transcription and data coding; (c) classroom observations, student interviews, transcription, and data coding; and (d) data analysis and triangulation.

### 5. Materials and Methods

Multiple data sources were used to provide a narrative into this Black homeschool collective to identify the mission, vision, and goals in the classroom experiences. These data sources were most important in helping to evaluate the presence of Afrocentric features across the curriculum, instructional methods, and student learning outcomes. Teaching in diverse classrooms requires an educator to understand the various literacies

that each student brings with then into the classroom. These literacies should be viewed as values, strengths, and assets to the learning environment. The BSA was chosen due to the homeschool collective's recognition as the only Black homeschool collective in the city and the only one of its kind in the local area. This school was the only school listed online as a collective and was also highly referenced by several local Black homeschool families.

This article explores how Yala attempted to promote familial relationships and the students' responses to those attempts. Each subsection explores how familial relationships were present at the BSA. In August 2019, two months prior to the beginning of the 2019–2020 school year, I conducted two formal interviews with Yala on two consecutive days—5 September 2019 and 6 September 2019—and interviews were approximately 1 to 1.5 h in length. I also spoke with Yala conversationally during classroom observations. During the 2019–2020 school year, I conducted six full days of observations for 1 month from October 2019 to November 2019 (specifically 9 October, 10 October, 16 October, 17 October, 7 November, and 8 November). I also scheduled three days—10 October, 17 October, and 8 November—to conduct interviews with four individual students that lasted approximately 40 min to 1 h. Classroom observations lasted four hours which was the approximate length of one full school day at this homeschool collective. Consequently, I present snapshots of brief classroom observations rather than an in-depth, full, or complete picture of daily life at the BSA.

## 6. Results

### 6.1. The Features and Benefits of Familial Relationships

According to teacher and student interviews and classroom observations, familial relationships, promoting an African time orientation and personalized learning plan in the teaching of *OurStory* and the practice of Rising Meeting, benefited students' academic and cultural lives. Getting to know your students through their interests and needs are important to the teacher's curriculum design and instructional practices, as lessons should be directly connected to students' lives. Yala employed familial relationships as a teaching style that can be applied in American schools due to its academic and cultural benefits. The teacher's teaching style consisted of familial relationships fostered by a village mentality and are suggested in the curriculum and instruction of teachers in American schools.

The employment of familial relationships led to Yala's implementation of an authoritative instructional approach, a prerequisite to establishing trust and a healthy rapport between the teacher and students. An authoritative instructional approach consisted of kinship connections, positive cultural affirmations, parental roles and responsibilities, and peer-as-sibling relationships. At the BSA, an authoritative instructional approach helped support a familial relationship between the teacher and students at this homeschool collective. Familial relationships were beneficial to the teacher and students as they developed personal relationships, engaged in open communication, and facilitated critical conversations.

Familial relationships, representative of an authoritative teaching approach, are important to the BSA as it provides a kinship environment that helps to promote a rapport between the teacher, students, and their families. Familial relationships also influence Yala's curriculum choices and instructional practices at the BSA. This rapport allows Yala to engage with and teach her students from an Afrocentric perspective that considers, addresses, and meets the students' educational and cultural needs and interests.

Yala's dual role as a teacher and family member helped students conform to a familial culture at this homeschool collective. One student, Wellington, often required additional encouragement and motivation to complete his class and homework assignments. Yala uses her teacher and familial identities to reinforce high and positive expectations and encourage students to persevere despite adversity through behavior that displayed love and affection at the BSA. For example, while some students struggled with assignments that were difficult for them, Yala took the necessary time to address the student's areas of concern and culturally affirm them to build their confidence and remind them that

they could achieve success. During the third classroom observation, on 16 October 2019, Wellington became upset about a class assignment with which he struggled. Yala rubbed his face soothingly, explained the task, and affirmed his potential for success. Yala explained to Wellington, "When I look at you, I see my son. I'm not just going to let you slip through the cracks . . . " The teacher's response to Wellington reveals that affection is used to emotionally connect with her students to help advocate for them in the classroom so that they feel accepted in a familial environment.

This verbal expression of Yala's familial connection with Wellington provides evidence of her affection towards students and how she culturally affirmed their value and abilities in the classroom. During an interview, Yala empathetically shared the importance of showing love, nurturance, and affection to students when she discussed the absence of these pillars of familial relationship in many public and private schools. She said: "At the end of the day, that's what I want, for them to know that they're loved, to know that they're great. That's the answer. That's the solution, period . . . " (Yala, interview, 6 September 2019) [48].

Due to Yala's cultural identity and understanding and respect of culturally specific hand gestures, she gives students fist bumps to show a cultural commonality between her and students. The colloquial expression of a fist bump, popularized as a greeting by President Barack Hussein Obama and his wife Michelle Obama, symbolizes approval or praise for the student.

Yala also uses familial roles as a surrogate parent to encourage students to develop a sense of respect for authority figures and elders and reinforce student expectations and positive behavior. Yala communicates her high expectations with students by employing an authoritative teaching style that helps promote a familial relationship reflective of a mother and child dynamic between the teacher and students, similarly found in a Black parenting style. The students understood Yala's redirection by conforming to the student expectations in the classroom. This relationship reinforced high expectations and warm demands reflective of an authoritative teaching style [49,51]. When students receive Yala's directive and corrective feedback, they respond by adhering to their teacher's advice as a form of respect for their familial relationship. Students respect their teachers when they exert control and high expectations [49,51]. This is obvious in the way that students receive and apply Yala's constructive feedback to improve their lives.

An authoritative instructional approach through positive reinforcement is not necessarily Afrocentric, but the way Yala practices the general principle, in conjunction with her familial relationships to students, promotes an Afrocentric vision. According to Walker (2009), "The best teachers were parents who provided increased support in response to failure and did not interfere with the child's autonomy after success. These teachers were identified as authoritative teachers" [52] (p. 125). An authoritative instructional approach is important to the Afrocentric classroom because it addresses the high expectations of the greatness of a student from a Black cultural perspective that places the needs and interests of people of African heritage at the center of the teaching and learning experience.

In addition, throughout the school day, Yala offers positive affirmations through verbal praise to students for answering questions correctly and helping their peers, reflecting a sibling relationship. For example, when a student answers a question correctly, she congratulates them with a response that reflects affection. During a math lesson, Yala praised a student's work ethic and offered encouragement when she said, "...Isis is on fire today! Be like this every day! I don't expect anything less!" Yala intends to use positive and culturally affirming words, phrases, and gestures to praise and encourage students' work ethic and independence to acquire their traditional greatness.

Authoritative teaching is one way that Yala intends to usher students to reach their fullest potential or what she refers to as "greatness". For example, during a classroom observation, Yala's celebratory responses affirmed students' "greatness". She was observed paraphrasing the following words to students at the end of each school day, "I see nothin' but greatness in you". Yala recognized students' effort, abilities, achievement, and "greatness", which speaks to the vision and goals of this homeschool collective. Yala

also recognized students' "greatness" with a certificate of recognition, equivalent to the "Student of the Week" acknowledgement often found in public and private schools. On the sixth and last classroom observation for this study, on 8 November 2019, Wellington was awarded for his academic progress. Although Wellington previously struggled with his confidence and academics, over two weeks, he showed improvement exhibited in his work ethic, independence, and motivation in completing class and homework assignments. As a result, Wellington was rewarded for his efforts with this certificate of recognition.

Yala promoted familial relationships that encouraged BSA students to develop a peer-as-sibling relationship. The students first resisted a kinship connection. However, the longer students attended this homeschool collective, the more their relationship and connection increased. This was apparent during classroom observations when a peer-to-sibling relationship and sharing of personal information in the classroom also existed among students to continue supporting other students' greatness.

Based on interviews and classroom observations, many BSA students did not initially agree with or accept the familial relationship that Yala encouraged due to their previous experiences at former public and private schools where kinship was practically non-existent. The BSA students only accepted the familial culture after spending some time at the homeschool collective where they became more comfortable with how Yala taught and interacted with students from a familial orientation. Familial relationships would become evident in students' behavior in the classroom particularly as they committed to motivating each other and conforming to Yala' high expectations during learning experiences.

Students need time and scaffolding to come around to the idea of peer-as-sibling relationships. Resistance ensued throughout the journey of students developing a peer-as-sibling relationship. Students like Isis and Wellington initially rejected the familial orientation at the BSA. According to an interview with Isis and classroom observations, she initially rejected a familial culture. For instance, Isis did not see her teacher or peers as a family as a new student at this homeschool collective. However, the familial culture at this homeschool collective influenced her to change and embrace kinship by the third classroom observation. Isis eventually encouraged and offered feedback to her peers as someone who cares about their development in the classroom. Isis encouraged a peer, Wellington, to follow directions and finish his class assignment when she clarified the teacher's instructions. During this classroom event, the school was a safe space to share personal information as a therapeutic teaching and learning moment.

On another occasion, peer-as-sibling relationships were particularly apparent between Isis and Wellington during classroom observations. When Isis took on the responsibility of an older student to motivate a younger peer, Wellington, to take school work seriously by turning in his homework assignments and paying attention in class, this motivation is an example of peer-to-peer encouragement through a familial relationship. This interaction was familial as it revealed the values in the classroom, including understanding, accountability, leadership, comradery, and community. Whether friendly or familial, a relationship was developed between both the teacher and across peers in their compassionate interactions, including helping one another, which indicated the development of a familial relationship between students.

Furthermore, Yala was always quick to praise and offer positive affirmations whenever she saw the students exhibiting this familial culture. When Yala noticed the familial relationship, she offered positive affirmation to encourage it to continue. For example, Yala was observed saying, "Way to help... we're family!" Yala consistently reiterated the importance of a familial relationship to establish a family-oriented culture among students at the BSA.

### 6.2. Personalized Learning Plan

Yala teaches generalized content to fulfill nationally and state-mandated core content standards expected of public and private schools and adhered to homeschooling laws. However, she also taught from a culture-infused curriculum consisting of historical and

cultural lessons focused on people of African heritage. As an Afrocentric educator, Yala's curricular choices and instructional practices revealed features of Afrocentricity including the implementation of an African time orientation and a personalized learning plan. This decision to teach beyond the mandated curriculum with a focus on Black history and culture speaks to Yala's investment in creating a familial environment with students and their families to accommodate students' developmental needs and cultural lives.

Yala's family-oriented curriculum was flexible in its consideration of students' academic and cultural needs. The flexibility schedule at this homeschool collective was noticed by many students at the BSA like Mia, a student who was pregnant, as positive and beneficial in stark contrast to previous schooling experiences. For example, Mia expressed that her previous public school teachers, at a predominantly White institution of teachers, administrators, and curriculum, often failed to assist her or other students with a question or offer academic support in the classroom. This was due to limited time or an inflexible school schedule that privileged a Western time orientation and the teaching of a mandated curriculum that focused on Europe and its people. Mia described her former public school and the way content was taught as "White everything", leading her, like so many other Black students, to feel "invisible" or not valued in the classroom.

In contrast to these previous negative experiences in public school, the Afrocentric aspects of the curriculum and instruction at the BSA reveal that Yala's flexible homeschool schedule directly addressed students' concerns and promoted their academic and cultural development. During another occasion, an African time orientation that was flexible and supportive was observed when Yala explained her high expectations for Wellington and his peers and gave him additional time to complete his unfinished homework assignment in class. The goal was for Wellington to take ownership of his education by holding himself accountable and completing his academic goals. Throughout classroom learning events, Yala adjusted and modified her teaching schedule to accommodate students. This respect for time was shown in the value for the production of quality student work through extra support and assistance.

The development of a personalized learning plan for students is another feature of Afrocentric education at this homeschool collective that reflects a familial relationship that is understanding of the students' needs for an individualized learning plan. A personalized learning plan is a benefit of a familial relationship established between the teacher and students. Yala supplemented this generalized curriculum with a personalized curriculum to fit both the academic and cultural needs of Black students. During an interview, Yala explained that there was not a "one size fits all" curriculum and that "You can't judge everybody according to the same standards". Each student had a personalized learning plan focusing on students' interests that were prioritized at the BSA. Yala explained during an interview that personalized student learning plans are "tailored to student learning, preferences, and interests". A personalized learning plan is one way that Yala attempts to build a familial relationship with students and their families. She acknowledges each student's personhood to foster strong and healthy connections. The development of a familial relationship between the teacher and students allowed Yala to adapt, adjust, and modify her curriculum and instruction to meet their needs. For example, during classroom observations, Yala emphasized student-centered and project-based learning opportunities, including pregnancy/parenting and Black entrepreneurship lessons.

In regard to pregnancy and parenting, one student, Mia, was interested in learning about and maintaining a healthy pregnancy, child rearing practices, and time management. Yala understood the importance of a healthy pregnancy. As a result, she provided Mia with books on what to expect as a new pregnant mother. Yala gifted Mia a popular best-selling book about pregnancy.

Black entrepreneurship is another feature of personalized learning found at the BSA. Yala references the *Nguzo Saba* or the Seven Principles of Kwanzaa, translated from Kiswahili. Using three *Kwanzaa* principles—*Nia* or purpose, *Ujamaa* or cooperative economics, and *Kujichagulia* or self-determination—Yala teaches entrepreneurial lessons em-

phasizing the importance of business ownership to address economic struggles and build wealth in the Black community. Yala connected her identity as an entrepreneur to the *Kwanzaa* principles to encourage the importance of financial literacy in the Black community. She described her financial and cultural responsibilities and independence when she said, "I don't get any grants, loans, corporate sponsors. I'm doing this for my ancestors. I know my *Nia*. I got *Kujichagulia*. I do this because of *Ujimaa*". Yala references the *Kwanzaa* principles to empower students to own a business that benefits the Black community which supports a familial or village orientation in the classroom.

Entrepreneurship lessons were also discussed in connection with *OurStory* lessons, applicable to students' lives and the real world. For example, Yala asked the students, "What's a vocation? Use your context clues". One student replied, "A job!" and Yala continued, "What's a collective?" When students do not answer, she replies, "Our collective vocation. It's like a group. It should be our job to build and develop our community". In this quote, Yala encourages students to reflect on their roles and responsibilities in the development of the Black community through entrepreneurship.

Familial relationships are evident in personalized learning plans with both academic and personal aspects. Yala provided an opportunity for Blair to learn how to start a business. This personalized learning plan developed from a familial relationship allowed Blair to adopt an entrepreneurial identity after acclimating to an Afrocentric education. According to classroom observations, Yala consulted with Blair and his mother to develop a personalized student learning plan focused on Black entrepreneurship consisting of books on economics and business management. This observation reveals a familial relationship between the teacher, student, and his mother, which further supported student-centered and customized assignments.

Due to his familial relationship with Yala, Blair was not only inspired by Yala's identity as a vegan and entrepreneur, but by the lessons that she taught on the Kwanzaa principles. During classroom observations, the Kwanzaa principles were often referenced to discuss the process and benefits of starting a business to build independence and wealth in the Black community which supports the village mentality in giving back to your neighborhood and the world. Based on observations, it was apparent that Blair was motivated by his teacher's identity and classroom lessons on the Kwanzaa principles to become vegan, start a business, name his business *Ujamaa Bakery* (a Kwanzaa principle), and provide vegan dessert products to the greater community. For example, the Kwanzaa principles are found in Blair's business name and purpose. According to an interview, by the end of the 2018–2019 school year, Blair started *Ujamaa Bakery*, a vegan dessert company with products sold in local farmer's markets and accessible to the Black community. The naming of his business revealed Blair's understanding and application of *Ujamaa* as a *Kwanzaa* principle to his personal life and company. Blair explained his inspiration for starting a vegan business when he said: "I learned about the Kwanzaa principles ever since I joined this school, and I started to learn more about it because we always recite an oath and a Kwanzaa principle every day". Blair further explained, "Ms. Yala gave me some motivation off it... She was talking about how in certain restaurants, they don't have any vegan food". Yala's encouragement of students' adoption of the *Kwanzaa* principles and an entrepreneurial identity in the classroom intends to build students' confidence, leadership, independence, and agency.

Blair's decision to start this business reflects Yala's influence. A familial relationship between the teacher and students allowed for a positive teaching and learning outcome that led students like Blair to develop an entrepreneurial identity. Yala's familial relationship with students like Blair is found in her attentiveness to students' interests. According to an interview with Blair, since he has attended the BSA, he has seen a drastic change in himself. He has become more interested in his education and starting a business. Blair, who explained, "I always said I wanted to create something, but I never did", broke free of this mentality as a homeschool student and began to identify as an entrepreneur.

Familial relationships are manifested through the teaching of *OurStory* and the practice of Rising Meeting. *OurStory* and Rising Meeting are two components of how familial relationships are used to help students build an understanding of their Black history and culture through accurate narratives and daily traditions. Both are beneficial components of familial relationships and venerate the contributions of African ancestors and encourage their purposeful legacy through the lives of their descendants who strive to accomplish the mission of liberation and restoration to one's traditional greatness.

*6.3. Teaching OurStory*

Familial relationships led to the teaching of *OurStory* which helped unearth untold truths and mistruths about Black history. There was a need to tell *OurStory* to teach students historical truths through authentic storytelling that connected historical issues to contemporary issues from a social justice lens. *OurStory* made history a personal exploration of one's heritage from a Black cultural perspective. BSA students who formerly attended public and private schools, expressed a disinterest in Black history. This disinterest was due to the absence of a focus on Black history and culture in previous schools. Many students were in disbelief as they learned untold truths about their history in previous public and private schools. Yala explained, "But once I showed [a former student] this, and then he started lookin', he came back to me, 'Ms. Yala, I'm sorry. You were right.' I said, 'See? I'm makin' progress,'" Yala described one student who questioned the lessons she taught but eventually believed her after researching the topic. According to Yala, this former student told her, "'I don't believe anything that you're saying'". However, when he fact-checked what Yala taught him, he became a believer in the truth. Yala explained, "That was another good day too. When they say, 'You're right, Ms. Yala'" (Yala, personal communication, 5 September 2019) [48]. Yala described another time when a student did not believe her: "When I told her this stuff, she didn't believe me! She was just shakin' her [head]—just in disbelief, in shock... A lot of my students don't believe me. They don't think they're great. One of 'em told me, 'Ms. Yala, you're lyin'.'" (Yala, personal communication, 6 September 2019) [48].

Due to a village mentality, students felt safe and supported during difficult conversations that encouraged them to unlearn mistruths about Black history and examine ways in which issues such as the mass incarceration of African American people are linked to a racist police force. Many students embraced truth telling and embodied a newfound love and respect for their Black history. During classroom observations, there was a focus on the content that the current teacher taught, her teaching approach, the delivery of the content, and student responses to the curriculum and instruction. A focus on critical conversations and the ways in which controversial or unlikely conversations were facilitated, scaffolded, and extended in the classroom among the teachers and students emerged from the data and were investigated as a benefit of a familial or village orientation in the classroom environment. These topics may not have been facilitated without a familial relationship between the teacher and students in the classroom.

Due to the familial relationship that was developed between the teacher and students, students challenged and unlearned inaccurate Black history and engaged in candid conversations about real-world issues impacting students throughout the teaching and learning experience, especially during *OurStory*, one of the most salient themes of familial relationships that Yala implemented and manifested at the BSA. Yala uniquely uses the term *OurStory*—the teaching of Black history from a Black cultural perspective—to name the curriculum and her approach to teaching content at this homeschool collective.

Familial relationships are present in two ways at the BSA: the name of the content and the learning experiences that emerge from the historical and contemporary lessons applicable to students' lives. First, familial relationships are present in the name of the content. *OurStory* is asserting a kinship fidelity and responsibility in the students. In this case, the name of the content, *OurStory*, signifies the cultural history of the teacher and students at the BSA. The emphasis on "Our" directly speaks to an accurate collective

history taught for and by Black people to "educate to liberate" and return students to their "traditional greatness". Hence, the title *OurStory* signifies a shared or common cultural heritage, ancestry, and lived experience among Black people or people of African descent.

Across many public and private schools, American history offers an inaccurate depiction of history and the Black experience in the United States. However, *OurStory*, in contrast to American history, offers an accurate account of history taught from a Black cultural perspective. At the BSA, Yala aims to prevent the miseducation of the Black student by teaching the truth from an Afrocentric perspective that challenges monocultural White American history that many of her students experienced as former public and private school students. *OurStory* or Black history lessons emphasized unsung and sung African American heroes and ancestors such as Mansa Musa, Granville T. Woods, and Harriet Tubman. Yala explains, "My problem with the school system is they teach our babies they come from slaves, they're descendants of slaves but what about Mansa Musa? The richest man ever!" (Yala, interview, 26 May 2017) [48].

Blair agreed with Yala when he said, "From being here, I learned about people I never knew like Mansa Musa, who is the richest man in history who passed along gold to other people... I was actually interested and surprised off of it". Yala invites students to "rule again" by using historical facts about the contribution of ancestors to apply to their lives. This concept of "ruling again" speaks to previous leadership and prestige inherent in African culture. The concept of "ruling again" is Afrocentric in the way that Yala "educates to liberate" and help students to return to their "traditional greatness" by presenting accurate narratives about Black history and culture.

Yala debunks historical myths with the *OurStory* curriculum by challenging traditional American history that often leaves out or devalues the presence of indigenous and Black persons. For example, in American history, myths are widely taught such as the mistruth that Black people are descendants of slaves rather than recognized first as African people and contributors to civilizations. A second mistruth is the myth that the Greeks and Romans are the architects of civilization. The third mistruth is the myth that Christopher Columbus, romanticized as an American hero, discovered America and helped the indigenous people on the continent. At the BSA, his exploitation of the indigenous Native American community is viewed as the beginning of the first incidence of human genocide and mass violence in America. These myths are far from the truth which is the motivation for Yala to tell the truth about history from a Black cultural perspective that is centered on people of African descent.

Yala's teaching of *OurStory* or Black history lessons led to the sharing of experiences during critical conversations and the contemporary impact of historical issues. During these conversations, the teacher and students connected institutional and systemic racial inequities that have impacted Black people throughout history to the present day. Topics included institutional racism, White privilege, racial violence, incarceration, and safety. These critical conversations connecting historical issues and current-day issues were a benefit of *OurStory*. Yala allowed students the time and space to share personal experiences as teachable moments in the classroom. The goal was to equip students to challenge issues, including institutional and systemic racism, and prepare students to solve real-world problems. These conversations illustrated a lively and passionate dialogue or back and forth transaction between the teacher and students. These conversations revealed students' interest in Black history that investigated, for example, the history of racism and the role of the fraternal order of police in connection to incidents of the unlawful police shooting of Black people throughout the U.S. Yala explained, "White supremacy is real. Our babies are taught that every day you walk out that door that you are a victim of White supremacy racism...racism is so subtle...our youth don't even think it exists any more..." (Yala, personal communication, 6 September 2019) [48].

Yala and students discussed unlawful police shootings four days after one particular police shooting. On 12 October 2019, Atatiana Jefferson was shot and killed inside her home by a White police officer during a non-emergency wellness check in Fort Worth, Texas [53].

Yala shared this recent news story to help students to see the value in the historical and contemporary issues of racial violence that have led to an outcry of protests for Black lives against racial profiling and police brutality impacting both Black men and women.

Yala encourages students to know about current events, especially issues impacting the lives of Black people. For example, during a classroom lesson, Yala connected the historical context of fugitive slave catchers to the fraternal order of police. She explained, "[Slave catchers] were the first police officers, so historically, that's really why our people don't like the police". This history of the fraternal order of police is discussed in the contemporary relevance of racial violence and the increase in police brutality cases leading to the unjust killing of unarmed Black people in America. Yala chooses to frame modern events in a historical way that connects to her mission and vision in liberating her students and helping them return to their traditional greatness. During this classroom observation, Wellington responded, "I don't like cops!" Yala used this current case to investigate the history of racism and challenge the criminality of Black people. Despite the racist history of the U.S. police, Yala presents a fair position as she challenges students to recognize two different perspectives, that not all police officers are "bad" or intentionally cause harm. Yala explained, "But wait a minute... At the end of the day, the police are to protect and serve... There are some good police officers". Yala is teaching students not to stereotype but to understand that not all police are racist despite the history of the fraternal order of police. A familial relationship helped the teacher and students to navigate these difficult conversations because trust was developed which allowed for a feeling of safety and comfortability.

This discussion encouraged many BSA students to talk about their personal experiences with police which extended to a conversation about the criminal justice system and issues of inequity. With a newfound understanding and respect for Black history or *OurStory*, students made personal connections to and challenged institutional and systemic racism on a historical and contemporary level connected to their lives and experiences. These personal discussions are grounded in an Afrocentric theoretical framework exemplifying the concept of familial relationships focusing on building a connection, rapport, and trust between the teacher and students. This relationship helped foster a comfortable, safe, and protective space for students to share true accounts of traumatic experiences. For example, many BSA students have been negatively affected by interactions with the police. During one classroom observation, Yala explained the importance of teaching *OurStory* in the classroom when she said:

BSA students expressed that they suffered from sleep deprivation and post-traumatic stress disorder (PTSD) after encounters with police officers. According to Love (2019), "Approximately 62 percent of all children come to school every day experiencing some type of trauma" [36] (p. 75). Based on student interviews, school violence was one of the primary motivations for an education at a homeschool collective.

One student, Isis, connected this discussion to a personal experience concerning her domestic living conditions and the previous incarceration of her parents. She explained, "My Daddy got locked up, and he's still doing good!... My momma got locked up—she's good". Wellington related to Isis when he said, "I know someone who got locked up". Students and teachers were engaged and connected to the conversation. Isis argued that even if previously incarcerated, people can change for the better, be successful, get their life back on track, and be "good". Isis referenced her parents as exemplars of people who were formerly incarcerated but managed to live decent lives after their release. This sharing led to a conversation about the unhealthy relationship that many police officers have with the Black community, including how the police have systematically targeted Black people as criminals. Yala explores the criminal justice system and police brutality that many Black people experience through civics.

Yala is aware of the impact of racist incidents on students' quality of safety and health and is concerned for her students' socioemotional well-being. During an interview, Yala explained, "By the time they get to me, sis, they so beat up, where they think they don't

have any options but the streets" (Yala, personal communication, 6 September 2019) [48]. Yala attempts to "educate to liberate" and return students to their "traditional greatness" by teaching Black cultural lessons to motivate them to value their education and, specifically, read above their grade level. During a classroom observation, Yala explained the importance of education, especially reading for Black boys. Yala read aloud a news article about the low reading levels of 8th-grade Black males on a national and local level. Without an established familial relationship between the teacher and students, this article could have been received wrongly or without a full understanding of motivating students to see the power and influence of education in combating pervasive racism, including the mass incarceration of Black boys which correlates with low reading scores. However, through familial relationships, the choice of the article demonstrates how the teacher and students are connected and responsible for educating to liberate.

Yala's familial approach to teaching is a method of racial protectionism as she is preparing students for a racist American society in the classroom of this homeschool collective. A benefit of familial relationships is that the teacher takes accountability and responsibility in preparing and protecting Black students who have been historically marginalized and disenfranchised. As a surrogate parent or parental figure, Yala attempts to prepare and protect students for a racist world that views Black people, especially men, as violent and criminals. Through these critical conversations and lessons, students become aware of systemic racism, allowing them to challenge it.

### 6.4. The Practice of Rising Meeting

Familial relationships are significant to the practice of Rising Meeting, a daily ritual at the beginning of the school day when Yala and students engage in culturally affirming recitations that lead into learning opportunities extending beyond Black history lessons to traditions that could be practiced daily. Rising Meeting comprises acknowledging the *Oath to Our Ancestors*, the visible symbol of the *African Flag*, and the *Nguzo Saba* or Seven Principles of *Kwanzaa* to build community at this homeschool collective. Students practice these physical artifacts and affirmations as a part of the vision and mission of the homeschool collective in achieving its objective to educate, liberate, and restore students to their traditional greatness. BSA students are not indoctrinated to a national identity but rather their African ancestry. The rituals at the BSA are symbolic of togetherness, cohesion, unity, and nationhood. Cultural artifacts that reflect your students' identities could also be brought into the classroom to show cultural pride in addition to having the American flag, map, pledge of allegiance, and any other symbol of America.

Rising Meeting is a morning ritual or daily practice at the beginning of the school day. At this time, the teacher and students participate in a community-centered recitation. This recitation interconnects with the teaching of *OurStory* and the recitation of the *Oath to Our Ancestors*. During an observation of Rising Meeting, Yala emphasized Black history, focusing on African ancestors and their contributions to the world and civilization.

During an observation of Rising Meeting, Yala taught Black history through storytelling and literature, focusing on African ancestors and their contributions to the world and civilization. She referenced the *Oath to Our Ancestors* which helped generate conversation about Black history. This Afrocentric poem and text culturally affirm, validate, and honor African ancestors for their sacrifices and contributions to the world. African people, recognized as inventors and intellectuals, are celebrated as the first contributors to civilization through their work in mathematics and science. Yala emphasized the contributions of African people to the building of pyramids and the study of astronomy. Black history lessons, interwoven throughout all aspects of this homeschool collective, reveal Afrocentricity.

During an observation of Rising Meeting, Yala affirmed the cultural greatness in her students by teaching the students the words and importance of the recitation which promotes the traditional greatness of African ancestors as a model of Black excellence. During the *Oath* recitation, Yala emphasized values in the poem, such as "Our African

Ancestors who brought science into the world" and "Our African Ancestors who brought civilization to the world". Yala uses the *Oath to Our Ancestors* to remind students of their "traditional greatness". During an observation, Yala explained, "Our ancestors were big on astronomy. What was the first calendar? The sun and the moon... Who started mathematics? Our African ancestors" (classroom observation, 9 October 2019) [48]. This *Oath* emphasizes the purpose of this homeschool collective in liberating and promoting greatness in students to recognize people who contributed to the world before they died, leaving a legacy for their descendants.

## 7. Discussion

While many public and private schools in the U.S. espouse a culture of low teacher expectations and low teacher morale, leading to the desire for independent Black schools, including Afrocentric schools [14,15,24], the Afrocentric educational strategy found in homeschooling has become a viable option to address the academic and cultural needs of Black students due to its familial orientation.

### 7.1. The Benefits of Familial Relationships

BSA students benefit from familial relationships in four ways: (a) establishing a joy for school; (b) improving academically and behaviorally; (c) developing self-confidence and self-worth; and (d) feeling safe and protected. Familial relationships are beneficial to supporting Black students' development of cultural pride, agency, self-determination, and independence.

### 7.2. Establishing a Joy for School

Yala helped students have an enjoyable learning experience through familial relationships, as expressed in the student interviews. Flexible and personalized lessons and assignments led students to having an increased interest in and enjoyment out of their education where there once was none. Mia explained that her homeschool experience was better than her experience as a former public school student. Since attending this homeschool collective, Mia has acquired a newfound interest in school. Mia explained, "I actually want to come to school. I remember I used to wake up like, 'I don't want to go.' I want to come to school now". Yala modified and adapted her content and how she taught to fit the needs and interests of her students. Yala taught content lessons using scaffolding and support that students needed and wanted to learn. This accommodation made for a fun and easy learning experience, as described by students.

Familial relationships between the teacher and students led students to have joy and an appreciation for their teacher and their education. BSA students contrasted their relationship with Yala and their teachers at their previous schools. They had a positive relationship with Yala, but more of a negative experience with teachers at their past public and private schools. Mia described her educational experience as positive due to her relationship with her teacher. She considered Yala one of her favorite teachers. Blair, another student, developed a relationship with his teacher and peers that he never had at his previous public school. He explained:

In the public school, I barely interacted with the teachers and the only time I interacted was when I had a question which I barely asked and here, I talk to Ms. Yala a lot more...I started to talk a lot more and started asking questions because I typically don't ask a lot of questions—only if I am curious about it. (Blair, personal communication, 10 October 2019) [48].

Blair's realization of enjoyment of school is at the crux of personal growth and academic confidence as he begins to build and maintain a positive relationship with his teacher while becoming more comfortable in social situations.

*7.3. Academic and Behavioral Improvements*

BSA students noticed a positive change in themselves including an improvement in their academics and behavior, including their confidence. According to Blair, while he attended public school, he struggled with math and science. Blair plans to continue his homeschool journey at home and at the BSA until graduation. He explained, "I was getting bad grades". Blair confessed to feeling more confident as a student at the BSA because of the community-oriented support from the teacher and other students. All BSA students expressed that they had experienced an increase in interest in their education and improvement in their academics. While some students experienced an academic change, other students like Mia experienced a behavioral and attitudinal change that her mother noticed.

Although there was an initial challenge when Yala presented Black history truthfully in a way that was different from the way students had previously learned, these lessons led students to become what Yala referred to as "believers" of *OurStory*. In a personal interview, Isis shared, "I didn't think [Yala] knew what she was talking about...We had a whole disagreement, and it took most of the class". Isis challenged Yala's teaching of *OurStory*, claiming that she was "lying".

Despite initial hesitancy, after students developed a familial relationship, rapport, and trusting relationship, they eventually believed the truthful lessons about Black history during *OurStory*. This connection helped students better understand the information that she taught about Black history. Students believed her teachings and refrained from questioning the validity of her Afrocentric lessons. Familial relationships, including affection and nurturing, made students believers not just in education or the Afrocentric classroom but in themselves and their future.

While BSA students initially challenged *OurStory* lessons, they eventually acquired a newfound knowledge and respect for Black history provided by Yala and believed her accurate teachings. Yala received acceptance from students or student buy-in. BSA students became interested in uncovering the truth about Black history and culture. One student, Isis, challenged the *OurStory* lessons that Yala taught. *OurStory* was unparalleled to the history that she learned while a former public and a private school student. This challenge was apparent during a classroom observation when Yala instructed the students to label ten countries in Africa on a map to demystify inaccuracies about Africa. However, Isis did not believe the lessons that Yala taught during *OurStory*. Yala explained during an interview that for many students, "This is their first time of hearing this, and it contradicts their whole paradigm". As students conformed to a familial culture at this homeschool collective, according to Yala, they became believers of the truth about Black history during the *OurStory* lessons and challenged previous inaccurate history lessons.

Students like Isis are learning more about the interconnection between their lives and the legacies of their African ancestors. This newfound knowledge about truthful Black history and cultural narratives derives from familial relationships as a benefit at this homeschool collective. Eventually, Isis expressed an appreciation for Black history taught from an Afrocentric perspective that values the true accounts of Black culture and life. During an interview, Isis confessed:

I like it more here because [Yala] tells us our part of the story as African American people instead of hearing it as the White people saying, 'Oh they were just slaves.' Like, 'No, there's more to it.' Instead of hearing one side of the story, you hear both sides. (Isis, personal communication, 8 November 2019) [48].

During an interview, Isis's new knowledge dispelled racist stereotypes, myths, and inaccuracies about African people and the continent. BSA students eventually became what Yala refers to as "believers", disrupting their miseducation at previous public and private school students. As a result, familial relationships at this homeschool collective helped students to excel both academically and behaviorally, thus, positively impacting their self-confidence and self-worth.

### 7.4. Developing Self-Confidence and Self-Worth

A familial orientation helps students build trust, rapport, community, and confidence with the teacher and their peers. As students learn more about their history and culture, they develop their character and are more apt to succeed. BSA students witnessed a change in their self-confidence and self-worth in developing a familial relationship with the teacher and their peers and a sense of belonging that they did not previously have. BSA students had negative experiences at their former public and private schools. Mia described feeling "invisible" as a former public school student. This experience negatively impacted her self-esteem. In contrast to this invisibility, the BSA is a homeschool collective where students' lives are valued and celebrated in the classroom through a familial orientation that considers their academic needs and cultural interests with a personalized learning plan. BSA students expressed seeing a change in themselves that included the development of cultural pride which boosted their self-confidence and self-worth. Another student, Isis, explained that after learning about Black history during *OurStory* lessons, it "Makes me feel proud because I know a lot about my African ancestors that I didn't before". Research has proven that Black students develop cultural pride when they learn about their history [38,41,42,50,54]. Yala explained, "The number one thing that I want them to take away—just to know their worth. Knowledge of self, and to know that they're great..." (Yala, personal communication, 6 September 2019) [48].

Yala's Afrocentric teaching methods and lessons have influenced BSA students' lives. According to student interviews, a familial relationship where other aspects of Afrocentric education were present led students to develop cultural pride, self-confidence, and self-worth. Blair spoke to this when he explained his increased self-confidence since attending this homeschool collective. Blair explained, "I felt like I didn't fit in at public school because I barely talked to people in class. I basically just sat and answered questions. Here, I'm able to make a lot more friends and be active". Another student, Wellington, also acknowledged a change in himself when he admitted to feeling a sense of belonging at the BSA that he had not felt previously at a public school. Wellington explained, "I feel like I fit in... I feel like people care about me".

Mia also struggled with feeling as though she did not belong while at her former public school. Feelings of insecurity and social anxiety were ramifications of previous schooling experiences. However, since attending the BSA, her confidence has drastically increased.

Another benefit of familial relationships was visible in the extended discussion of critical conversations that emerged from *OurStory* lessons. These lessons revealed an increase in students' confidence in knowing the truth about their cultural history.

### 7.5. A Safe and Protective Space

According to interviews, many BSA students experienced school violence such as peer bullying, school fights, and racism at previous public and private schools. However, once they attended the BSA, they were not confronted with this issue or concern. An established familial relationship between the teacher and students was the difference. This relationship offered a safe and protective learning environment for students that were not reflective of BSA students' previous experiences with school violence in any of its manifestations. During an interview, Mia explained, "Here, [bullying] honestly wouldn't really happen". Yala alluded to the nature of the familial relationship and how it supports a kinship-oriented environment. Afrocentric schools, such as the BSA, promote non-violence that supports the academic and cultural development of Black students. As intended, a familial relationship teaches students that school is not a place for violence or "negative sanctions for belittling, humiliating and embarrassing others" [55] (p. 235).

Essentially, Yala took on the role of surrogate parent, caregiver, teacher, and counselor as found in the literature on Black teachers [56]. During interviews, Yala described herself in the classroom as a "Momma. Counselor. Principal. Dean... A child of one is a child of all—that's an African proverb. So, I look at them as my children" (Yala, interview,

6 September 2019) [48]. Parental roles explored teachers' descriptions of their responsibilities and interactions with students at this homeschool collective. Familial relationships were beneficial to the teachers and students as they developed personal relationships, engaged in open communication, and facilitated critical conversations.

First, a familial relationship is beneficial because it allows teachers to get to know their students on a personal level that extends beyond the academic level, builds a rapport with students, and addresses concerns that are normally shunned in traditional classrooms for an emotional connection. Second, a familial relationship is beneficial because it encourages open communication between teachers and students, promoting a safe, healthy, and nurturing relationship.

Third, family relationships are beneficial because they allow students and teachers to have candid and critical conversations on various topics. According to teacher interviews, school violence, including racism, is the primary reason Black families are homeschooling their children. This is accurate at the BSA. A familial relationship encourages a safe and non-violent environment. As a result, this familial relationship is believed to be free of school violence, including peer bullying and racism. This is evident in teachers' discussion of critical conversations that are often important to Black families in raising their children to survive in a racist world. Across Black households, Black parents must teach their children how to respond to police officers as a form of protection for Black students to become well-equipped to challenge and resist racism and survive in a racist society. This is a method of *racial protectionism* [16,18,26].

There are many pathways towards Black resistance through Black radical educational politics that encourage the leadership to confront White supremacy in curricula, instructional practices, and spaces. The school environment is an ecosystem and repository that students of diverse backgrounds have access to many times more than they are home for a day. They spend hours upon hours in school, which speaks to its primary influence on students' lives. There is a dire need for Black radicalism in education to exist in schools to promote a pro-Black curriculum, praxis, and learning experiences that support, for example, critical race theory. Educators must learn from Afrocentric scholars who uplift Black scholarship, narratives, and perspectives. Adopting an Afrocentric perspective makes visible the value and significance of the Black cultural perspective across politics in ways that allow for progressive change and educational reform in the classroom.

It is my hope that this paper will help empower diverse educators to create responsible, accountable, and transparent educators in a collectivistic environment that disrupts an individualistically dominated society as a worldview across industries, including education. We must create the spaces in which our students are able to survive and thrive. Educators must recognize that they are a part of the village of family members, mentors, and role models that instill academic, cultural, and social values in students. We need more village mentality-minded educators in our schools to take up arms just as homeschooling parents are already doing.

## 8. Conclusions

While there are many types of schools that offer models of success between teachers and students, my study focuses on one Afrocentric teacher. Yala's teaching style provides lessons for teachers in American public and private schools as a means for fighting for a high-quality and equitable education for students. Culturally diverse students need access to this level of curriculum.

Familial and kinship connection is nurturing, supportive, and reflective of the Seven Principles of *Kwanzaa*, including *ujamaa*. Ujamaa, defined by its traditional African origin in Swahili as "familyhood", "extended family", or as a family-oriented principle, is a social and economic policy implemented by President Julius Nyerere of Tanzania [57]. The concept of *ubuntu*, which is the understanding that a person develops through the influence of the community as they share the same belief in this paraphrased African proverb [58],

was understood by all three teachers who described their interactions with students as familial through their use of kinship terms, for example.

However, some of the familial approaches found at the BSA that are not mentioned in this article are not recommended for American public and private schools due to the nature of the homeschool. However, this article focuses on the teaching techniques that Yala used that can be implemented in a culturally diverse classroom in American public and private schools.

While this article highlights the historical issues concerning public and private schools in the U.S., everyone's experience is different. This article does not seek to portray public and private schools as entirely problematic for Black students. It also does not romanticize Afrocentric education, as it is not always the best alternative for students nor one of the many ways to educate Black students successfully. Today, this debate of the most effective strategies or the best way to teach Black students is still ongoing.

This article recommends that teachers use a culturally inclusive curriculum and instruction, such as Afrocentric education, to educate culturally diverse students, especially Black students, in all educational contexts. The benefits and best practices of an Afrocentric education in the ways Black students learn and Black teachers instruct are racially protective in promoting cultural pride, agency, self-determination, independence, liberation through education, and a return to their traditional greatness, as considered in this article, thus, further informing the education field. Afrocentric features in curricula and instruction are beneficial to students in any educational setting, whether public, private, parochial, charter, or homeschool.

**Funding:** This research received no external funding.

**Institutional Review Board Statement:** The study was conducted in accordance with the Declaration of Helsinki, and approved by the Institutional Review Board (or Ethics Committee) of University of Louisville (protocol code 17.0466 and 8/9/2019) for studies involving humans.

**Informed Consent Statement:** Informed consent was obtained from all subjects involved in the study.

**Conflicts of Interest:** The author declares no conflict of interest.

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
