# Peer review of "Afrocentric Education for Liberation in the Classroom: It Takes a Village to Raise a Child"

_education, doi:10.3390/educsci13060532_

Round 1

Reviewer 1 Report

This is a significant study, and the authors should be applauded for highlighting the view that it takes a village to raise a child especially within the context of racial equity. 

The introduction required a little more with regard to external citations in order to spotlight the gap in this area. 

Statistics have been used effectively in the initial section. 

I did think that the literature review required extension especially with regard to spotlighting the research gap. 

More information should have been sought about home schooling of students of color, in relation to racism or racial equity within schools. The researchers should ask the question of with their students are being home-schooled on account of prejudice and discrimination at school. 

I was deeply appreciative of the Afrocentric theoretical framework coaching formed the study. This should be thoroughly referenced.

Familial relationships are evidently a significant feature of home-schooled education, especially within the collectivist communities and village-oriented education. 

The Afro-centric theoretical framework required considerable external support - the researchers will need to draw on theories and change initiatives within this context in order to justify its use. This will allow the paper to be more academically rigorous. 

The studying is well constructed over a four month period, and consists of interviews observations and personal contact. 

A significant area requiring work would be drawing on other research in the field to support the assertions of this manuscript. 

The research findings are quite telling and will be of great value to racial equity in contemporary schools. 

Better organization of the Results section would be helpful especially since large chunks of material/findings are presented. 

The Discussion would benefit from further external citations and other like studies - there are very few studies cited in the section which undermines its credibility. 

It is important that the authors include a section on implications for practice - this will be of value to students and teachers in mainstream education - the fundamental purpose of this manuscript appears to be directed at change so the inclusion of implications will be vital 

It is important to return to the Afrocentric educational context highlighted earlier. 

The manuscript makes an important contribution to educational equity in contemporary classrooms however it does require some refinement before it is publication-ready. 

Reviewer 2 Report

This paper makes an interesting, examining Afrocentric practices, particularly “familial relationships through culturally responsive instructional practices” which engages “a personalized learning plan, authoritative teaching, OurStory, and Rising Meeting”.  It is rightly enthused that such instructional practices do indeed benefit Black students and enhance their educational experiences. The paper’s strength lies in its case study approach to examining the Black Scholars Academy (BSA), a Black homeschool collective in the Midwest of the United States.  The truth is we need more of such empirical studies of Afrocentric educational/instructional practices as “one of many culturally responsive techniques to best teach culturally diverse students, especially Black students, in educational settings”. For one thing such studies help us debunk dominant narratives of ‘lack of hard evidence’ on the success of Afrocentric teachings for Black students.  The author’s engagement with Afrocentricity as ‘best practice’ in a homeschool collective has relevance not only for Black students but for all students, including those even outside of “Pre-Kindergarten (Pre-K) to 12th-grade homeschool collective”. Educational innovations must learn from pluralistic contexts that also utilize diverse teaching, instructional, pedagogic knowledge base for learners.

Clearly the paper is worth publishing but I suggest some areas for strengthening in order to reach a broader appeal on the question of how we can rethink Black educational futurities:

First, on Research Methodology: the focus on “six full days of classroom observations, individual interviews with the research participants for in-depth analysis”, critical examination of study materials - books and content on the BSA home school collective’s website, and interpretation of the research findings through “an evaluation of textual artifacts, including teacher instruction” provides a wealth of details.  There are questions of how do we allow the use of such multiple data sources to bounce off each other? How do we draw the different implications relating to teaching diversity even within the Black students’ home school collective – [i.e., questions around gender, sexuality, geography, religion, class, region and nationalities?] These questions are worth taking up for their practical and philosophical relevance in addressing critical issues that strengthen Afrocentric pedagogies, and the often-misguided critiques of dominant scholars on what Afrocentricity is about. Furthermore, why the BSA was chosen and what questions were asked and why this need to tell “our story”.

Second, the use of the Afrocentric theoretical framework to inform the study of the Black homeschool collective where the teacher teaches from an Afrocentric lens is apropos. In explicating on framework, the author must engage the question of African spirituality and its connections to ideas espoused in the principles of Kwanza for example. This area of theorization needs strengthening because Afrocentricity is not just a method, but a philosophy of practice and epistemology of educational advocacy and resistance.

Third, while I enjoyed reading the sections on: Familial relations, Black joy, self-confidence and worth, etc., I would like to see the author explore further the question of pathways of Black resistance through Black radical educational politics that help today’s learners confront White supremacy and its logics, and schools are carceral places.  Relatedly, in the current climate of White Fascisms and its ‘anti-woke fantasies’ it will help to embrace the connections of Afrocentric thought with critical race theory. Afrocentric scholars need to draw on this linkage, as unfortunately some Black scholars see a divergence.  Their reluctance to speak about this connection or convergence is not only a ‘Black betrayal’ of our Ancestors and their rich intellectual heritage, but also, a woeful failure of the part of the contemporary African scholar to work with the true meaning of a powerful idea: ‘to know is to act politically and responsibly’.

Fourth, I am looking at the literature the author is drawing on. While I am aware the paper is speaking to a US context, there are parallels and convergences in Black education challenges globally and drawing on other literature outside the US context will be an added strength. It definitely moves the discussion away from being a purely US-centric analysis.

Fifth, the paper must highlight and complicate what is meant by educational ‘success’’. The Afrocentric paradigm brings a different meaning that extends success beyond ‘academic’ to social’, as well as a confluence of educators, learners, home and parental responsibilities.

Sixth, I have often argued that in North American contexts, people often misunderstand the African adage ‘It takes a Village to Raise a Child”. It is not simple that it is a family/group or community that raises a child! What the adage is speaking to is the fact that we [as educators and a society] must work to create/have communities in the first place with shared responsibilities, transparencies and accountabilities to each other. What are the broader implications of this paper on this for schooling and education in general?

Finally, [unless I mis it] I would like to see a brief discussion of the author/s subject location fronted in the essay. I know sometimes writers do not want to do that. But, I believe this is essential. The personal subject[ive] location is critical in terms of what brings the author to the examination of this topic. It calls for an analytical contextualization of writer/self as a methodological and discursive feature of the discussion. Such personal subject location helps the reader to understand the perspective from which one is conducting the discussion/analysis. By perspective, I do not mean just ideology or analytical framework, but a personal accounting of why the author writes about what she/he does philosophically and politically perhaps.

Round 2

Reviewer 1 Report

In the first research question, avoid using an acronym.

Thank you for the prompt attention to the suggested changes.  
